# High STI burden among a cohort of adolescents aged 12–19 years in a youth-friendly clinic in South Africa

Matt A. Price[1,2], Monica Kuteesa[3], Matthew Oladimeji[4], William Brumskine[4,5], Vinodh Edward[4,5], Heeran Makkan[4], Funeka Mthembu[4], Vincent Muturi-Kioi[3], Candice Chetty-Makkan[6], Pholo Maenetje[4,5]*

1 IAVI (Formerly International AIDS Vaccine Initiative), New York, NY, United States of America, 2 University of California at San Francisco, San Francisco, California, United States of America, 3 IAVI, Nairobi, Kenya, 4 The Aurum Institute, Rustenburg, South Africa, 5 Department of Medicine, Vanderbilt University, Nashville, Tennessee, United States of America, 6 Health Economics and Epidemiology Research Office, Wits Health Consortium, Faculty of Health Sciences, University of Witwatersrand, Johannesburg, South Africa

☯ These authors contributed equally to this work.
* PMaenetje@auruminstitute.org

**Data Availability Statement:** Anonymized data are available at Figshare: https://doi.org/10.25382/iavi.25491859.v1.

## Abstract

Adolescents face a higher risk for HIV, STIs, and unintended pregnancy than any other age group in sub–Saharan Africa, and have unique health care needs as they navigate this period of growth and developmental milestones. We conducted the Youth Friendly Services study among adolescents in Rustenburg, South Africa to address some of these concerns. Participants aged 12–19 were followed quarterly for 12 months, asked at baseline about demographics, their sexual behavior, and tested for HIV, STIs, and pregnancy (girls). Report of sexual activity was not a requirement for enrollment. Assent and parental consent were obtained for participants under 18. Some follow up visits fell during COVID-mandated shut-downs, and we worked with participants to reschedule and extend follow up as appropriate. Here we present data on reported behaviors, participant attrition, risk of HIV, other STI, and pregnancy. From May 2018 to August 2019, we enrolled 223 HIV-negative, non-pregnant adolescents (64% girls). The median age was 17 (IQR: 14–18). Among the 119 (53%) participants who reported being sexually active at baseline, the median age at first sex was 16 years (IQR: 15–17). During follow-up, an additional 16 (7%) participants reported having their first sexual encounter. Among the sexually active participants, the incidence of HIV was 1.5 cases / 100 person-years at risk (PYAR, 95% CI: 0.4–6.0), the incidence of chlamydia was 15.7 cases (95% CI: 10.1–24.4), gonorrhea was 4.7 cases (95% CI: 2.1–10.5), and HSV was 6.3 cases (95% CI: 3.1–12.6); we observed no cases of incident syphilis. The incidence of pregnancy among sexually active girls was 15.0 pregnancies / 100 PYAR (95% CI: 8.5–26.5). Despite small numbers, the incidence of most STIs was significantly higher in females compared to males. We also observed two pregnancies and 5 incident STIs among participants who reported never having had sex, these tended to be younger participants. From March to September 2020, the clinic was shut down for COVID-19, and 53 study visits were postponed. Follow up was concluded in November 2020, a total of 19 participants were lost to follow up, however only one participant dropped off-study during COVID-19

**Funding:** This work was funded by IAVI and made possible by the support of many donors, including United States Agency for International Development (USAID). The full list of IAVI donors is available at http://www.iavi.org. The funder had no role in the study design, data collection and analysis, decision to publish, or preparation of the manuscript. The contents of this manuscript are the responsibility of the authors and do not necessarily reflect the views of USAID or the US Government.

**Competing interests:** The authors declare that no competing interests exist.

shutdowns. Retention at the final visit was 91.5%. We successfully completed a prospective study of adolescents to learn more about the risks they face as they navigate sexual debut in the context of a program of youth-friendly counseling and services. Among self-reported sexually active participants, we observed a high rate of HIV, STI and pregnancy, however we also observed pregnancy and STIs among those who reported no sexual activity.

## Introduction

Adolescents are a critical target population for sexually transmitted infection (STI) interventions, unplanned pregnancy prevention, and long-acting HIV prevention products including HIV vaccines and broadly neutralizing antibodies [1–3]. In 2020, 150,000 new HIV infections were recorded among adolescents aged 10–19 years; In sub–Saharan Africa, less than 25% of adolescents have received HIV test results, and six of every seven new infections among those aged 15–19 are in girls [4]. STI and unplanned pregnancy remain significant problems with high rates among adolescents in Africa, the burden of this problem often falling heavily on adolescent girls and young women [4–6]. The SARS CoV2 pandemic has exacerbated the situation, as testing and service interruptions have accounted for drops in HIV testing and referrals for HIV care during this time [7, 8].

The landscape of HIV prevention technologies to increase choice, utilization, satisfaction, and effectiveness for young people is rapidly changing. Results from HPTN 083 and HPTN 084 have shown that long acting, injectable prep (Cabotegravir) can prevent HIV acquisition, and evidence for the Dapivirine Vaginal Ring has led to its approval in several African countries for HIV prevention [9]. New studies to examine these modalities in adolescents are underway or recently completed [10–12]. Previous studies including simulated vaccine trials (i.e., "mock" trials using a licensed product but otherwise designed to mimic the rigors of a clinical trial) have also shown that while adolescents are willing to participate in clinical research [13–16], concerns about issues such as vaccine seropositivity, logistical issues, community engagement, and parental/guardian permissions remain [17]. In addition, inadequate access to adolescent friendly healthcare services remain problematic, including tailored screening for sexual risk and addressing low perception of risk [18, 19]. These factors may not only impact enrollment and retention in clinical research but may also affect counselling during clinical trials as well as health care delivery.

IAVI and the Aurum Institute are both nonprofit institutions with similar missions of supporting and translating scientific discoveries into affordable public health solutions for those who need them most. Integral to this mission is understanding the needs of those for whom we develop products, in this case adolescents. The aim of this study is to determine the feasibility and acceptability of recruiting and enrolling adolescents, boys and girls aged 12–19, within an adolescent friendly clinical trial research setting in Rustenburg, South Africa. In this manuscript, we describe the study population and present data on retention, incident HIV, pregnancy, and STIs.

## Methods

We conducted a 12-month prospective observational cohort study with both qualitative and quantitative elements. Here we present the quantitative data.

## Study setting and population

This study was conducted at the Aurum Institute's Clinical Research site in Rustenburg, Northwest Province. Rustenburg is an urban town with high HIV rates, and a well-developed health-care infrastructure. It has an approximate population of 562,031 in 2022, with 25% being 15 years old or younger [20]. The economy is depending on mining, with the platinum mining industry being a major source of employment. The study was conducted among local 12-19-year-old, HIV-negative, non-pregnant female and male adolescents.

Prior to initiating the research, stakeholders' meetings were held with staff from the Department of Education, youth related organizations, and the Department of Social Development and Department of Health to discuss the study and recruitment. The study team also recruited potential participants at places where youth gather, youth groups, youth centers, and other points of community convergence. Adolescents were also reached through their parents via outreach recruitment leaflets, door to door visits and using social media platforms. The outreach teams would include a parent of a potential volunteer who would help to mobilise parents by leading study education sessions in the community. From there, interested parents would be registered in the recruitment dataset and scheduled for a clinic visit where study procedures took place. Parents or guardians were contacted directly for recruitment of adolescents (ages 12–17). Recruitment was supported by adolescent community advisory group. The participants primarily came from 8 areas (Monakato, Paadekraal, Tlhabane, Boitekong, Geelhout, Ramochana, Ikageng, Maile) in Rustenburg.

Potentially eligible participants were invited to attend the research clinic for additional details and enrollment. Participants were eligible for enrollment into the cohort if they were aged 12 to 19 years; were HIV antibody negative by rapid test; agreed to the study procedures including giving consent and assent, providing locator information, completing questionnaires, undergoing physical examination, collection of blood and genital samples at 3-monthly intervals for testing of HIV and other STI; and had a negative pregnancy test (females). Sexual activity was not a requirement for enrollment.

All adolescents attending the clinic for study screening were offered HIV voluntary counseling and testing, health-education counselling, free condoms, and free access to the general care clinic for the total duration of the project, whether enrolled into the cohort study or not.

## Adolescent friendly services

We adopted and implemented at least five of the ten National Adolescent-Friendly Clinic Initiative (NAFCI) standards required for Adolescent and Youth Friendly Services (AYFS) recognition, as set out by the South African Department of Health (DoH) [21, 22] and the WHO [23]. These included: (i) management system support for the effective provision of adolescent and youth friendly health services; (ii) appropriate adolescent health services are available and accessible; (iii) the clinic has a physical environment conducive to the provision of adolescent friendly health services; (iv) provision of relevant information, education and communication (IEC) promoting behavior change and consistent with the YFS essential service package; and that (v) the clinic provides continuity of care for adolescents and that proper referral systems are in place. In collaboration with the DoH, we conducted assessments to ensure adherence to these standards and adequacy of services. We also implemented a participant satisfaction survey to obtain additional insights about health services provided, and revised services per responses received.

## Enrolment and follow up procedures

Participants typically attended five or six visits: screening, enrolment (typically within one week of screening), and follow-up visits at 3 months, 6 months, 9 months, and 12 months after

enrollment. Screening and enrollment sometimes took place on the same date. Participants were invited to come to the clinic whenever they experienced STI symptoms or other health problems. Parents / guardians were invited to attend one visit and selected participants (adolescents or parents/guardians, in separate groups) were invited to attend focus group discussions; these data are not presented here.

Following the informed consent process, participants were then interviewed using a structured questionnaire that was administered face to face by trained study staff. At each visit, we collected information on demographics, mental health, physical health, social networks, HIV treatment experience, health-related quality of life, and sexual behavior. At each visit, laboratory testing was conducted as described below. We conducted a full physical examination at baseline, with symptoms-directed (e.g., report of STI) examinations at follow up visits. All participants with HIV infection had their CD4-count and viral load measured, and were referred to an accredited HIV-care center for care and antiretroviral therapy. From 26 March to 20 September 2020, due to the COVID-19 pandemic, clinic visits were halted due to a national shutdown. We worked with participants to reschedule follow up visits, extending follow up as appropriate.

## Laboratory procedures

Serum specimens were tested for antibodies against HIV-1 (Abbott Determine HIV-1/2 with confirmation by UniGold, Trinity Biotech). Assays for other STIs included HSV-2 (IgG ELISA test, Kalon Biological Ltd., Guildford, UK) and syphilis (RPR confirmed by IMMUTREP TPHA, OMEGA Diagnostics Limited, Scotland, UK); *Neisseria gonorrhoeae* and *Chlamydia trachomatis* were diagnosed on endocervical specimens (females) or urine (males) using the Cepheid gene expert CT/NG Rapid PCR Test (Cepheid, Sunnyvale CA, USA). To test for pregnancy, we used QuickVue One-Step HCG Urine Pregnancy Tests (Quidel Quickvue; Quidel Corporation, San Diego, CA). All laboratory testing for these infections was performed at the Rustenburg research clinic's laboratory.

## Statistical analysis

**Basic analyses, incidence, and predictors of study dropout.** Data were analyzed using STATA 16.0 (Stata Inc., College Station, TX, USA). Study endpoints included incident HIV infection, incident non-HIV STIs, incident pregnancy, and study drop out. We describe participants baseline characteristics and reported behaviors using proportions and means, and compared by sex using the appropriate statistical tests. Because we observed cases of STI and pregnancy even among those participants who do not report sexual activity, where we present STI and pregnancy data we report using two different denominators: 1) participants who report sexual activity (i.e., report being at risk for STI, pregnancy) and 2) all participants (i.e., including those who do not report sexual activity).

We calculated the incidence of new cases of HIV, STIs, and pregnancy (females) overall and stratified by participant sex. Date of HIV/STI infection or pregnancy was estimated as the midpoint between the last negative and first positive test result. Rates for each outcome were determined by dividing the number of cases over the person years at risk (PYAR). The PYAR were calculated as the sum of the time on study while reporting sexual activity (report of sexual activity was not required for enrollment). Sexual activity was recorded as yes or no to the question "Are you or have you been sexually active?", and participants were asked to consider as sexual activity, acts of vaginal intercourse (defined as when the penis is put inside the vagina), anal intercourse (defined as when the penis is put inside the anus/butt/bum), and oral intercourse (defined as when the mouth touches the penis, vagina, or anus). For those who reported

initiating sexual activity while on study, the start of their PYAR was assumed to be the mid-point between the date of first reporting sexual activity and the date from the prior visit when they reported no sexual activity. Drop out is shown as overall percentage of participants who were lost to follow up, and calculated as a rate, similar to the methods above. Lost to follow up was defined as a person who did not attend their final study visit, and the date of their last attended study visit was considered the date they went off study. Persons who were lost to follow up immediately after enrollment (i.e., never returned for follow up) were assumed to have contributed one day of study participation.

### Ethical considerations

To participate in the study, written assent was required from adolescents aged 12–17 years along with their parents/guardians' consent, and written consent was required from those aged 18 or 19. A Setswana version of the information sheets and consent forms were available to participants whenever necessary to facilitate comprehension as Tswana is the most common language in the area. The study was approved by the Human Research Ethics Committee of the University of Witwatersrand in South Africa (WHREC Reference no. 170607).

## Results

### Study cohort

We conducted enrollment from 17 January 2018 to 15 August 2019, and follow up ended on 26 November 2020. Of the 237 adolescents screened, 223 (94%) were enrolled. The reasons for failed screening included pregnancy (n = 6), unavailable for full follow up time (3), not willing to use contraception (2), HIV seropositive (2), and one each of: not willing or able to provide informed consent, not willing to agree to blood collection, and too old at the time of enrollment.

More adolescent girls (64%) than adolescent boys (36%) were enrolled, and the median age of enrolled participants was 17 years (interquartile range (IQR) 14–18, Table 1). Participants were allowed to select 'transgender/transexual' as a response to gender, none selected this option. About 99% of participants self-reported their race as "African," with two (1%) participants reporting race as "coloured". Most were long term residents of the community, with a median of 15 years (IQR: 12–18) living in the study community; only 14 (6%) reported living in the community for a year or less. Most (64%) attained standard 8 (grade 10) education or above and females tended to report significantly more education than males (Table 1). Almost a tenth (9%) reported recreational drug use in the past year, compared to nearly two thirds reporting alcohol use (60%) in the same period; reported drug use was significantly more prevalent among males than females (Table 1). About half (119, 53%) reported sexual activity at enrollment, an additional 16 (7%) reported their first sexual encounter during study follow up. More than half of the males reported being circumcised (58%) and a minority of females reported a previous pregnancy (17%), all of whom reported only one prior pregnancy.

### Reported sexual behavior, HIV, STIs, and pregnancy

A total of 135 (61%) participants reported sexual activity (i.e., ever participating in vaginal, anal, or oral sex), 119 of whom reported activity at enrollment. Among those reporting sexual activity, the median age of sexual debut was 16 years (IQR: 15–17), six participants could not remember the age at which they first had sex. Participants were asked about same-sex sexual activity, none was reported. Most participants reported that all their sex partners were within five years of their age, and report of older sex partners was significantly more common among

**Table 1. Baseline characteristics of adolescents enrolled in a longitudinal study in Aurum, Rustenburg, South Africa, stratified by participant sex.**

| | Total | Male | Female | p-value |
|---|---|---|---|---|
| | N = 223 | N = 80 | N = 143 | |
| Age at enrollment (median, IQR) | 17.0 (14.0–18.0) | 16.0 (14.0–19.0) | 18.0 (14.0–18.0) | 0.66 |
| Age at enrollment (years) | | | | 0.064 |
| 12–16 | 96 (43.0%) | 41 (51.2%) | 55 (38.5%) | |
| 17–19 | 127 (57.0%) | 39 (48.8%) | 88 (61.5%) | |
| What language is spoken at home? | | | | 0.092 |
| Tswana | 192 (86.1%) | 64 (80.0%) | 128 (89.5%) | |
| Xhosa | 16 (7.2%) | 7 (8.8%) | 9 (6.3%) | |
| Other* | 15 (6.7%) | 9 (11.3%) | 6 (4.2%) | |
| What's the highest level of education you've completed? | | | | 0.012 |
| Standard 7 or below | 78 (35.0%) | 37 (46.3%) | 41 (28.7%) | |
| Standard 8–10 | 142 (63.7%) | 41 (51.2%) | 101 (70.6%) | |
| Other** | 3 (1.3%) | 2 (2.5%) | 1 (0.7%) | |
| What do you do for a job? | | | | 0.97 |
| Un- or self-employed∞ | 36 (16.1%) | 13 (16.3%) | 23 (16.1%) | |
| Student | 187 (83.9%) | 67 (83.8%) | 120 (83.9%) | |
| How would you describe your relationship status? | | | | 0.12 |
| Single with steady partner | 103 (46.2%) | 30 (37.5%) | 73 (51.0%) | |
| Single with casual partner(s) | 14 (6.3%) | 7 (8.8%) | 7 (4.9%) | |
| Single with no partner | 106 (47.5%) | 43 (53.8%) | 63 (44.1%) | |
| Reported recreational drug use, past year | | | | <0.001 |
| No | 203 (91.0%) | 65 (81.3%) | 138 (96.5%) | |
| Yes | 20 (9.0%) | 15 (18.8%) | 5 (3.5%) | |
| Reported alcohol use, past year | | | | 0.84 |
| No | 90 (40.4%) | 33 (41.3%) | 57 (39.9%) | |
| Yes | 133 (59.6%) | 47 (58.8%) | 86 (60.1%) | |
| Among those who report drinking (n = 133), how frequently do you drink? | | | | 0.55 |
| Less than once a month | 59 (44.4%) | 21 (44.7%) | 38 (44.2%) | |
| About once a month | 41 (30.8%) | 16 (34.0%) | 25 (29.1%) | |
| About 2–3 times a month | 27 (20.3%) | 9 (19.2%) | 18 (20.9%) | |
| Once a week | 4 (3.0%) | 0 (0%) | 4 (4.7%) | |
| More than once a week | 1 (0.8%) | 0 (0%) | 1 (1.2%) | |
| Missing | 1 (0.8%) | 1 (2.1%) | 0 (0%) | |
| Report of sexual activity, ever ‡ | | | | 0.89 |
| At enrollment | 119 (53.4%) | 41 (51.2%) | 78 (54.5%) | |
| During follow up | 16 (7.2%) | 6 (7.5%) | 10 (7.0%) | |
| None reported | 88 (39.5%) | 33 (41.3%) | 55 (38.5%) | |
| Self-report of circumcision (males) | | | | NA |
| No | 32 (40.0%) | 32 (40.0%) | | |
| Yes˚ | 46 (57.5%) | 46 (57.5%) | | |
| Missing | 2 (2.5%) | 2 (2.5%) | | |
| Self-report of previous pregnancy (females, sexually active at enrollment) | | | | NA |
| No | 63 (80.8%) | | 63 (80.8%) | |
| Yes | 13 (16.7%) | | 13 (16.7%) | |

*(Continued)*

**Table 1.** (Continued)

|  | Total | Male | Female | p-value |
|---|---|---|---|---|
|  | N = 223 | N = 80 | N = 143 |  |
| Missing | 2 (2.6%) |  | 2 (2.6%) |  |

\* Other languages include Tsonga (6), Sotho (3), Portuguese (2), Swati (1), Zulu (1), English (1), and Shona (1)

\*\*Other education includes Diploma(s)/ Occupational Certificates (2), adult school (1)

∞Only one participant reported self-employment, the remainder (35) reported being unemployed and out of school

‡ Sexual activity included report of vaginal, anal, or oral sex (see methods)

˚Circumcision includes one cultural circumcision at baseline, not performed by a doctor or nurse

NA: Not applicable (only one sex reporting data)

females than males (Table 2). One 17-year-old boy reported having a female sex partner between 6–10 years younger than he was. Only three (2%) participants reported knowing a partner was HIV positive, two reported a single HIV+ partner, one reported two HIV+ partners. Over half, 75 (55%), reported always using condoms. However, eight (11%) of those 75 who reported consistent condom use later reported a recent incident where they hadn't used a condom, but wished they had (Table 2). About 13% reported previously having an STI (Table 2).

Overall, during the study, 67 (30%) participants had 137 visits where an STI was detected. At baseline there were 40 participants with prevalent STI, including four participants with coinfections (3 cases of HSV and chlamydia co-infection, one case of gonorrhea and chlamydia). Prevalent chlamydia was significantly more common among females than males (Table 3). Three prevalent STIs were detected among participants who reported never having had sex (Table 3). HIV and pregnancy were criteria for screen out at enrollment, thus there

**Table 2. Sexual behavior reported at baseline (visit sex first reported) among 137 adolescents reporting sexual activity.**

|  | Total | Male | Female | p-value |
|---|---|---|---|---|
|  | N = 137 | N = 50 | N = 87 |  |
| Total number of lifetime sex partners | 2.0 (1.0–3.0) | 3.0 (1.0–5.0) | 2.0 (1.0–3.0) | 0.081 |
| Report of sex partner >5 years older |  |  |  | 0.003 |
| No | 119 (86.9%) | 49 (98.0%) | 70 (80.5%) |  |
| Yes | 18 (13.1%) | 1 (2.0%) | 17 (19.5%) |  |
| How often did you use condoms with your sex partners? |  |  |  | 0.19 |
| Never | 13 (9.5%) | 5 (10.0%) | 8 (9.2%) |  |
| Sometimes | 49 (35.8%) | 13 (26.0%) | 36 (41.4%) |  |
| Always | 75 (54.7%) | 32 (64.0%) | 43 (49.4%) |  |
| In the last 3 months, did you ever not use a condom, but you thought you should have? |  |  |  | 0.11 |
| No | 32 (23.4%) | 9 (18.0%) | 23 (26.4%) |  |
| Yes | 33 (24.1%) | 11 (22.0%) | 22 (25.3%) |  |
| Not sure | 5 (3.6%) | 0 (0.0%) | 5 (5.7%) |  |
| NA, always used condoms | 67 (48.9%) | 30 (60.0%) | 37 (42.5%) |  |
| Were any of your partners living with HIV? |  |  |  | 0.91 |
| No | 134 (97.8%) | 49 (98.0%) | 85 (97.7%) |  |
| Yes | 3 (2.2%) | 1 (2.0%) | 2 (2.3%) |  |
| Report of previous STI |  |  |  | 0.18 |
| No | 119 (86.9%) | 46 (92.0%) | 73 (83.9%) |  |
| Yes | 18 (13.1%) | 4 (8.0%) | 14 (16.1%) |  |

**Table 3. Prevalence of STIs at enrollment among participants who report sexual activity (n = 119) and everyone (n = 223), comparing males vs. females.**

| | Total | Male | Female | |
| --- | --- | --- | --- | --- |
| | N (%) | N (%) | N (%) | p-value* |
| Any STI ** | | | | |
| Reporting sex | 37 (31.1%) | 8 (19.5%) | 29 (37.2%) | 0.04 |
| Everyone | 41 (18.4%) | 9 (11.2%) | 32 (22.4%) | 0.05 |
| Gonorrhea test results | | | | |
| Reporting sex | 6 (5.0%) | 1 (2.4%) | 5 (6.4%) | 0.32 |
| Everyone | 6 (2.7%) | 1 (1.3%) | 5 (3.5%) | 0.35 |
| Chlamydia test results | | | | |
| Reporting sex | 22 (18.5%) | 4 (9.8%) | 18 (23.1%) | 0.08 |
| Everyone | 23 (10.3%) | 4 (5.0%) | 19 (13.3%) | 0.05 |
| HSV test results | | | | |
| Reporting sex | 13 (10.9%) | 3 (7.3%) | 10 (12.8%) | 0.36 |
| Everyone | 15 (6.7%) | 4 (5.0%) | 11 (7.7%) | 0.44 |

*Pearson's chi-square test comparing prevalence by participant sex

** Combining Gonorrhea, Chlamydia, and HSV test results

were no "prevalent" pregnancies, or cases of HIV. During follow up, 36 participants had incident STIs including HIV (Table 4). This included two reinfections during follow up (both chlamydia with multiple negative chlamydia tests between the two positive results) and 7 coinfections (four chlamydia and HSV coinfections, and one each of: gonorrhea and chlamydia, HIV and HSV, and HIV and chlamydia). The incidence rate for HIV was 1.5 cases/ 100 PYAR

**Table 4. Incidence of HIV, STIs, and pregnancy in study participants who report sex during study participation (n = 135) and overall (n = 223) comparing males vs. females.**

| | | | | | Male | | | | Female | | | | |
| --- | --- | --- | --- | --- | --- | --- | --- | --- | --- | --- | --- | --- | --- |
| | Total cases | PYAR | Incidence* | 95% CI | Cases | MYAR | Incidence | 95% CI | Cases | FYAR | Incidence | 95% CI | p-value** |
| Incident HIV | | | | | | | | | | | | | |
| Reporting sex | 2 | 133.4 | 1.5 | 0.4–6.0 | 0 | 48.8 | 0 | – | 2 | 83.9 | 2.4 | 0.6–9.5 | 0.94 |
| Everyone | 2 | 223.4 | 0.9 | 0.2–3.6 | 0 | 81.8 | 0 | – | 2 | 141.6 | 1.4 | 0.4–5.6 | 0.95 |
| Incident Gonorrhea | | | | | | | | | | | | | |
| Reporting sex | 6 | 131.7 | 4.6 | 2.0–10.1 | 0 | 48.8 | 0 | – | 6 | 82.9 | 7.2 | 3.3–16.1 | 0.06 |
| Everyone | 7 | 222.4 | 3.1 | 1.5–6.6 | 0 | 81.8 | 0 | – | 7 | 140.6 | 5.0 | 2.4–10.4 | 0.04 |
| Incident Chlamydia | | | | | | | | | | | | | |
| Reporting sex | 20 | 134.1 | 14.9 | 9.6–23.1 | 3 | 48.8 | 6.1 | 2.0–19.1 | 17 | 85.3 | 19.9 | 12.4–32.0 | 0.04 |
| Everyone | 22 | 224.8 | 9.8 | 6.4–14.9 | 3 | 81.9 | 3.7 | 1.2–11.4 | 19 | 143.0 | 13.3 | 8.5–20.8 | 0.02 |
| Incident HSV | | | | | | | | | | | | | |
| Reporting sex | 8 | 133.3 | 6.0 | 3.0–12.0 | 0 | 48.8 | 0 | – | 8 | 84.5 | 9.5 | 4.7–18.9 | 0.03 |
| Everyone | 10 | 224.1 | 4.5 | 2.4–8.3 | 0 | 81.8 | 0 | – | 10 | 142.3 | 7.0 | 3.8–13.1 | 0.01 |
| Incident Pregnancy | | | | | | | | | | | | | |
| Reporting sex | NA | – | – | – | NA | – | – | – | 12 | 84.7 | 14.2 | 8.0–25.0 | NA |
| Everyone | NA | – | – | – | NA | – | – | – | 14 | 142.6 | 9.8 | 5.8–16.6 | NA |

PYAR: Person years at risk, MYAR: Male years at risk, FYAR: Female years at risk

*Incidence reported as cases per 100 years at risk and 95% confidence interval (CI)

**Incidence rate ratio comparing males to females within each row

NA: Not applicable

(95% CI: 0.4–6.0) however both cases of incidence HIV were among females; the incidence rate for HIV among females only was 2.4 (95% CI: 0.6–9.5) (Table 4). The incidence for each individual STI is shown in Table 4 and tended to be significantly higher among females. There were no cases of incident gonorrhea among sexually active males, but due to small numbers the gender-based difference only approached significance (p = 0.06). Chlamydia incidence also varied significantly by sex, with females being more than three times as likely to be chlamydia positive over study follow up (incidence rate ratio 3.2, 95% CI: 0.9–17.3, p = 0.04). Like gonorrhea, we observed no incident cases of HSV in males, and 8 cases among females (p = 0.03). There were 12 incident pregnancies among participants who reported sexual activity, and the incidence of pregnancy was 14.2 pregnancies / 100 FYAR (95% CI: 8.0–25.0).

Sexual activity was inconsistently reported and did not always correlate with STI. We observed 67 participants who reported sexual activity (50% of all participants reporting any sexual activity), but then recanted their reports at one or more later visits, claiming they had never engaged in sexual activity. For the purposes of this analysis, once a participant had reported sexual activity, they contributed PYAR until they completed their participation in the study. We observed two pregnancies, one case of gonorrhea, three cases of chlamydia (one prevalent at baseline), and four cases of HSV (two prevalent cases) among participants reporting that they were not sexually active at all. These participants are not initially counted in our calculated incidence rates as they do not contribute PYAR; prevalence and incidence including those reporting and not reporting sexual activity is shown in Tables 3 and 4. The two participants with a positive pregnancy test without reported sex were both 13 years old, two of those with positive chlamydia test results without reported sex were 12, the third was 17; the participant with a positive gonorrhea result in the absence of reported sexual activity was 14. Those with positive HSV results without reported sexual activity were aged 16, and 17 at enrollment. One participant who was HSV positive was male, the remainder were female. Age and sex were not significantly associated with STI, pregnancy and unreported sexual activity (data not shown).

### Study dropout

Among the 223 participants, 19 (8.5%) dropped out of the study, including three who never returned for any follow-up after enrollment, and one whose initial missed follow up visit fell during the COVID-19 related clinic shut down (late March through September 2020) and did not return for subsequent follow up. Fifty-eight study visits for 55 participants fell during this shutdown. The overall retention at study end was 91.5%, with an attrition rate of 8.4 participants dropped out per 100 person-years on study (95% CI: 4.6–12.1).

### Discussion

In this study to provide adolescent-friendly health care and services to a cohort of adolescents not recruited on report of "high risk" behavior, we observed a high prevalence and incidence of sexually transmitted diseases, even among those participants who did not report sexual activity. For all non-HIV STIs, rates were higher among females compared to their male counterparts. Retention in the study was high, despite clinic closures during COVID-mandated shutdowns. Dozens of study visits needed to be rescheduled, however only one of the 19 study dropouts were lost during the time the clinic was shut down. While the unprecedented events unfolding under the global COVID-19 pandemic makes it challenging to put our STI (including HIV), pregnancy, and retention estimates in context, many have observed high rates of STI and unplanned pregnancy in clinical research [5, 6] and our retention rates seem good in light of the pandemic and the challenges inherent in recruiting and retaining adolescents

[24, 25]. Our work further highlights the challenges in getting accurate data on sexual behavior [26–28], as we noted inconsistent and implausible self-reported behavior.

STI incidence in sub–Saharan Africa, particularly among adolescent girls, remains high. Our STI incidence was also high, and we recruited our study participants from youth organizations and local schools, without screening for "high risk" behavior. Even with relatively small numbers we see higher rates of STI among enrolled adolescent girls compared to their male study counterparts. In work done with at-risk 16–25 year old females as part of HPTN 082, over half (55%) had an STI detected, with the most common being chlamydia (27.8 cases/100WY) followed by gonorrhea with a rate of 11.4 cases / 100 WY [29]. In a study in Kenya with sexually inexperienced females aged 16–20, nearly three quarters reported sex with 56% of sexually active participants experiencing an STI (defined as infection with chlamydia, gonorrhea, trichomonas vaginalis, or HSV-2) during study follow up, nearly half of whom experienced multiple STIs. The authors report that within one year of first sex, one quarter of participants had an incident STI; similar to our study, chlamydia was the most common STI [30]. The pregnancy rates we observed were similar to other rates in female sex workers in Kenya (11.3 pregnancies/100WY in Kilifi and 17.9 pregnancies in Nairobi) and adolescent girls and young women in Cape Town who were not recruited based on report of higher risk sex (13.7 pregnancies/100 WY) [31]. A more recent cohort study in females aged 14–24 who report sex work in Kampala, Uganda, observed a higher incidence of unplanned pregnancies of 23.5 pregnancies/100WY [32]. A recent review of HIV incidence among adolescent girls and young women in sub Saharan Africa also finds that our HIV incidence estimates compare to their results, from 51 studies finding results ranging from a low of 0.4 cases/100 WY in Uganda to a high of 7.8 and 8.6 cases/100 WY among females aged 15–19 and 20–24 in KwaZulu-Natal South Africa, respectively [33]. In two cohort studies our team conducted in Rustenburg prior to this work, we observed an HIV incidence of 3.0 cases/100WY among women aged 18–35 years [34], and 9.5 cases/100WY in a subsequent smaller study of at risk women [35].

Frequently, published work on adolescent sexual and reproductive health is done with older adolescents, typically greater than 15 or 16 years of age, but we observed several cases of pregnancy and STI among much younger adolescents, including those who did not report sexual activity. Some studies have suggested that younger age may be associated with mis-reporting sexual activity data [28, 36] and while our average age at sexual debut was 16, it is clear from our data that high risk sex is happening in adolescents as young as 12 (our lower limit of age for this study). Although we had too few participants for robust testing of differences with age, two of our fourteen observed pregnancies were in girls who did not report sexual activity and both were thirteen years old, and two girls with chlamydia infection detected while not reporting sexual activity were twelve years old. Creating a safe and trusting environment for youth to encourage truthful disclosure of sexual activity remains a significant challenge, working with younger adolescents should be considered in the future as there are few published studies of sexual and reproductive health in adolescents under the age of 15 [37].

While high rates of unplanned pregnancies and STIs remain a problem, improving data collection for reliable and valid indicators of sexual activity also poses a challenge for research and public health. In our study, we observed prevalent STIs among enrolling participants who reported never having engaged in sexual behavior, which we defined as vaginal, oral, or anal sex. Other discrepancies in reported behavior we recorded included incident cases of pregnancy and STIs among participants who report never having engaged in sexual behavior; over 10% of "all the time" condom users later confiding that they had not used condoms during sex when they felt they should have; and participants who had reported sex later refuting this, claiming they had never engaged in sexual activity when asked at a later study visit. Challenges with recording sexual behavior in young people are common. In a study mentioned above

following sexually inexperienced adolescent girls in Kenya, the authors found that those who failed to report initiating sexual activity were more than three times as likely to contract an STI than their counterparts who did report sex [30]. A study in Jamaica in young women found increasing inconsistencies over time when comparing reported condom use to vaginal swabs positive for PSA indicating unsafe sex, suggesting that misreporting of condomless sex may have increased from enrollment to follow up, which the authors suggest may be related to changes in counseling messages and social desirability bias [27]. Another study in Kenya looked at two specific inconsistencies in reporting sexual behavior, what the authors termed "reborn virgins" and inconsistent report of timing of sexual debut. They found that one in five adolescents provided inconsistent data, including 8% who claimed no sexual activity after initially reporting being sexually active, and 12% who provided conflicting answers to the year of their sexual debut, which the authors concede may be due to not remembering correctly rather than any social desirability bias [36]. We also observed a few adolescents who could not or did not report their age at sexual debut, though we only measured this at first report of sex and could not compare data with another timepoint. This issue is not limited to adolescents. In a study of married, adult heterosexual couples where one partner has HIV and the other doesn't, the authors found that a third of all observed pregnancies, nearly half of observed sperm-positive vaginal samples, and 58% (M+F-) or 39% (M-F+) of linked HIV-transmission events happened during follow up intervals where the couples reported only using condoms [38]. Sexual behavior is hard to measure properly.

Our study did have good retention. With one year of follow up and extensions allowed due to COVID-related interruptions of service, we were able to retain over 90% of our study cohort. Study retention is critical for randomized critical trials, and our retention compares favorably to recent trials in similar populations. A recent Ebola vaccine trial reported a 95% retention at one year in their adolescent participants [39]. Adolescents can be a challenging group to engage and retain, and at the other extreme, a recent trial in Kenyan high school students only retained 54% of participants after the 7-month study concluded [40]. The strength of our cohort retention may in part be due to our efforts to engender an adolescent friendly environment, youth community advisory board, transportation of study participants to and from the site, convenient operating hours, and ensuring parental buy-in during the adolescent enrollment visit.

We implemented adolescent friendly services to accommodate adolescent unique needs, promote provision of comprehensive and quality health care, facilitate access and uptake of health care services, and enhance cohort retention. We observed that provision of health information/education tailored for youth and appropriate referral systems were both rated as highly satisfactory by study participants (data not shown). On the other hand, implementing adolescent friendly services in the context of a research study was not without its challenges–in particular keeping study visits short was particularly hard. This was in part due to lengthy study procedures, but also due to long waiting-times between and before procedures.

Our study had a few limitations. We did not employ methods to confirm sexual activity, such as testing vaginal samples for y chromosomal DNA in sperm or PSA, or measuring PrEP concentrations in biological samples. Some computer-administered survey instruments may afford some sense of privacy or confidence; we did not use these technologies during study follow up visits. During the consent procedure, parents were invited. After the informed consent process, adolescents and parents participated in study activities separately, parents were not present when adolescents were asked questions about their behavior. Adolescent participants were informed that everything they shared was confidential, and that their parents would not be informed of anything they shared. However, some adolescents may have been reluctant to confide accurate sexual behavior data with parents on site, even if they were not present during

the interview. Our eligibility criteria were set up to mimic those that might be found in a clinical trial (e.g., HIV uninfected participants, females who are not pregnant), and these study results should therefore be generalized to other populations with caution. We had hoped to enroll approximately equivalent numbers of boys and girls, however more girls reported for screening, and ultimately to complete enrollment, we enrolled more girls than boys.

## Conclusion

We successfully enrolled and followed cohort of South African adolescents, and our retention appears to have been little impacted by the COVID-mandated service shutdowns in 2020, as our study came to a close. The high rates of STDs among adolescents not recruited for "high risk behavior" shows an ongoing strong need for intervention. Self-report of sexual behavior remains problematic, investigators and public health officials should remain vigilant, and consider interventions across age groups, regardless of reported sexual activity.

## Acknowledgments

We thank the staff of the Youth Friendly Services clinic for their help and dedication to improving the health and wellbeing of South African adolescents.

## Author Contributions

**Conceptualization:** Matt A. Price, Monica Kuteesa, William Brumskine, Vinodh Edward, Heeran Makkan, Funeka Mthembu, Vincent Muturi-Kioi, Candice Chetty-Makkan, Pholo Maenetje.

**Data curation:** Matt A. Price, Matthew Oladimeji, Vinodh Edward, Heeran Makkan, Funeka Mthembu, Candice Chetty-Makkan, Pholo Maenetje.

**Formal analysis:** Matt A. Price, Matthew Oladimeji, Heeran Makkan, Pholo Maenetje.

**Funding acquisition:** Vinodh Edward, Vincent Muturi-Kioi, Candice Chetty-Makkan, Pholo Maenetje.

**Investigation:** Monica Kuteesa, William Brumskine, Vinodh Edward, Heeran Makkan, Funeka Mthembu, Vincent Muturi-Kioi, Candice Chetty-Makkan, Pholo Maenetje.

**Methodology:** Matt A. Price, Monica Kuteesa, William Brumskine, Vinodh Edward, Heeran Makkan, Vincent Muturi-Kioi, Candice Chetty-Makkan, Pholo Maenetje.

**Project administration:** Monica Kuteesa, William Brumskine, Vinodh Edward, Heeran Makkan, Funeka Mthembu, Candice Chetty-Makkan, Pholo Maenetje.

**Resources:** William Brumskine, Vinodh Edward, Heeran Makkan, Funeka Mthembu, Vincent Muturi-Kioi, Candice Chetty-Makkan, Pholo Maenetje.

**Supervision:** Matt A. Price, Monica Kuteesa, William Brumskine, Vinodh Edward, Heeran Makkan, Funeka Mthembu, Candice Chetty-Makkan, Pholo Maenetje.

**Validation:** Matt A. Price, William Brumskine, Vinodh Edward, Heeran Makkan, Funeka Mthembu.

**Writing – original draft:** Matt A. Price, Monica Kuteesa, Matthew Oladimeji.

**Writing – review & editing:** Matt A. Price, Monica Kuteesa, Matthew Oladimeji, William Brumskine, Vinodh Edward, Heeran Makkan, Funeka Mthembu, Vincent Muturi-Kioi, Candice Chetty-Makkan, Pholo Maenetje.

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
