## [Decision Letter · Decision Letter 0]

4 Jan 2024

PONE-D-23-24060High STI burden among a cohort of adolescents aged 12-19 years in a youth-friendly clinic in South AfricaPLOS ONE

Dear Dr. Price,

Thank you for submitting your manuscript to PLOS ONE. After careful consideration, we feel that it has merit but does not fully meet PLOS ONE’s publication criteria as it currently stands. Therefore, we invite you to submit a revised version of the manuscript that addresses the points raised during the review process.

We look forward to receiving your revised manuscript.

Kind regards,

Tinashe Mudzviti, MPhil(MD)

Academic Editor

PLOS ONE

Journal Requirements:

“This work was funded by IAVI and made possible by the support of many donors, including United States Agency for International Development (USAID). The full list of IAVI donors is available at http://www.iavi.org. The funder had no role in the study design, data collection and analysis, decision to publish, or preparation of the manuscript. The contents of this manuscript are the responsibility of the authors and do not necessarily reflect the views of USAID or the US Government.”

“This work was funded by IAVI and made possible by the support of many donors, including United States Agency for International Development (USAID). The full list of IAVI donors is available at http://www.iavi.org.  The funder had no role in the study design, data collection and analysis, decision to publish, or preparation of the manuscript.  The contents of this manuscript are the responsibility of the authors and do not necessarily reflect the views of USAID or the US Government.”

“This work was funded by IAVI and made possible by the support of many donors, including United States Agency for International Development (USAID). The full list of IAVI donors is available at http://www.iavi.org.  The funder had no role in the study design, data collection and analysis, decision to publish, or preparation of the manuscript.  The contents of this manuscript are the responsibility of the authors and do not necessarily reflect the views of USAID or the US Government.”

Additional Editor Comments (if provided):

Major comments:

1. The title of the paper and the objectives described are not sufficiently relatable. The title has a focus on STI burden whilst the aim described in the paper is looking at the feasibility and acceptability of recruiting and enrolling adolescents. The results presented are thus unrelated to the objectives set and thus cannot be thoroughly evaluated.

2. Authors must clearly describe the study setting. Initially (line 76) this is defined as, " Aurum Institute’s Clinical Research site in Rustenburg, Northwest Province." However, line 82-83 then lists additional study sites.

3. By restricting participants eligible to enrol for the study (pregnant, not willing to use contraceptive, diagnosis of HIV) the external validity of the results obtained is significantly reduced. Authors must then define to which population these results are generalizable.

Minor comments

1. The sentences in line 46 - 47 need references and those in line 48 need references that are more recent and not from 2018 (these are already 5 years old)

2. line 64: Define IAVI first before using the acronym.

3. line 125: should read, viral "load"

4. Please be consistent with terminology. The phrases, "young people, young women or young men" are not interchangeable with the word, "adolescent."

Reviewers' comments:

Reviewer's Responses to Questions

**Comments to the Author**

1. Is the manuscript technically sound, and do the data support the conclusions?

Reviewer #1: Yes

Reviewer #2: Yes

2. Has the statistical analysis been performed appropriately and rigorously? 

Reviewer #1: Yes

Reviewer #2: Yes

3. Have the authors made all data underlying the findings in their manuscript fully available?

Reviewer #1: Yes

Reviewer #2: Yes

4. Is the manuscript presented in an intelligible fashion and written in standard English?

Reviewer #1: Yes

Reviewer #2: No

5. Review Comments to the Author

Reviewer #1: The authors report on the high prevalence and incidence of sexually transmitted infections among adolescents from one clinic in South Africa. The authors also show that with the provision of youth friendly services, enrolment and retention of adolescents in studies related to sexual health can be very good. I only have a few minor comments.

Can the authors comment on the reasons why there were more female than male participants. Also discuss the reasons why females had a higher prevalence and incidence of STIs than males.

Can you also describe in more detail what makes the setting youth-friendly?

Lines 46-47 add reference to the statement

Line 58 clarify what is meant by simulated vaccine trials

Line 76 can the authors provide some data with regards to the catchment population size for the clinic/ adolescent population size

Line 119 Given that adolescents may be reluctant to talk about issues relating to sexual health with their parents, can the authors comment on how this was handled in the study?

Line 133: Neisseria and Chlamydia are organism names and should be capitalised and written in italics

Table 3: please also include any STI as a category

Reviewer #2: In this prospective observational cohort study, the authors report a high prevalence of incident sexually transmitted infections and pregnancy in adolescents in Rustenburg, South Africa. The authors describe an adolescent friendly service, effective methods of adolescent recruitment and high retention in this hard-to-reach population.

The data is well presented and tables are clear. The findings support previous literature reporting high rates of STIs in adolescents in Sub-Saharan Africa.

My main concern is regarding the general style of the scientific writing throughout the paper, it is often informal and inconcise. The first sentences of the discussion and conclusion are clear examples. Table headings should be reviewed and shortened (e.g table 3 ‘among everyone overall’) , use of phrases such as ‘on the other hand’, ‘in a study mentioned above’ should be avoided.

These data are important but may be better suited to an STI focused journal, the authors are advised to review the scientific writing style.

6. PLOS authors have the option to publish the peer review history of their article (what does this mean?). If published, this will include your full peer review and any attached files.

Reviewer #1: **Yes: **Ioana D Olaru

Reviewer #2: No

---

## [Author Response · Author response to Decision Letter 0]

2 Jun 2024

Journal requirements, Reviewers’ comments and authors’ replies: 

Journal Requirements:

1. Please ensure that your manuscript meets PLOS ONE's style requirements, including those for file naming. …

Author reply: Thank you for these links, we have updated a few elements of our manuscript to match these requirements. 

2. Did you know that depositing data in a repository is associated with up to a 25% citation advantage ...

Author reply: While we appreciate this advantage, we enrolled adolescents, and as part of the consenting process we made clear that protecting data and participant anonymity was very important. Since this is a protected population and sensitive data were collected, we will not be able to make these data freely available. 

“This work was funded by IAVI and made possible by the support of many donors, including United States Agency for International Development (USAID). The full list of IAVI donors is available at https://url.avanan.click/v2/___http://www.iavi.org___.YXAzOmlhdmk6YTpvOjRjOWRmZTI2ZWU3MTI1YjIxNjA2OGY0N2IwNzcxZWY5OjY6NWUyMjo3ZjVkZmE5ZGI4NjA2MzNjMzliZjhlZTYxOTJiMTJiZWJiOWIzZjUxYzhjN2YwNWZmMTFiNjdjYmMyNDlhOTE1OnQ6VA. The funder had no role in the study design, data collection and analysis, decision to publish, or preparation of the manuscript. The contents of this manuscript are the responsibility of the authors and do not necessarily reflect the views of USAID or the US Government.”

Please provide an amended statement that declares *all* the funding or sources of support (whether external or internal to your organization) received during this study, as detailed online in our guide for authors at https://url.avanan.click/v2/___http://journals.plos.org/plosone/s/submit-now.___.YXAzOmlhdmk6YTpvOjRjOWRmZTI2ZWU3MTI1YjIxNjA2OGY0N2IwNzcxZWY5OjY6NjVlMDpiMzM1ZjQwNzcyNDI5NzJlMzA4Y2QyMTQyOWE1NmZhYzFmZjJlNDI4MjIzNTUxZTk0MzUyODAxY2I5MTU1ODUwOnQ6VA Please also include the statement “There was no additional external funding received for this study.” in your updated Funding Statement.

Author reply: Thank you for your comments on our funding and acknowledgements statement. We have revised them as follows: 

Acknowledgements

We thank the staff of the Youth Friendly Services clinic for their help and dedication to improving the health and wellbeing of South African adolescents. 

Funding Statement

This work was funded by IAVI and made possible by the support of many donors, including United States Agency for International Development (USAID). The full list of IAVI donors is available at http://www.iavi.org. The funder had no role in the study design, data collection and analysis, decision to publish, or preparation of the manuscript. The contents of this manuscript are the responsibility of the authors and do not necessarily reflect the views of USAID or the US Government. We also wish to acknowledge the support from the University of California, San Francisco’s International Traineeships in AIDS Prevention Studies (ITAPS), U.S. NIMH, R25MH123256. The content is solely the responsibility of the authors and does not necessarily represent the official views of the National Institutes of Health. There was no additional external funding received for this study. 

“This work was funded by IAVI and made possible by the support of many donors, including United States Agency for International Development (USAID). The full list of IAVI donors is available at http://www.iavi.org. The funder had no role in the study design, data collection and analysis, decision to publish, or preparation of the manuscript. The contents of this manuscript are the responsibility of the authors and do not necessarily reflect the views of USAID or the US Government.”

“This work was funded by IAVI and made possible by the support of many donors, including United States Agency for International Development (USAID). The full list of IAVI donors is available at https://url.avanan.click/v2/___http://www.iavi.org.___.YXAzOmlhdmk6YTpvOjRjOWRmZTI2ZWU3MTI1YjIxNjA2OGY0N2IwNzcxZWY5OjY6NDRmNDozYWMwMDc4ZjA5ZDRlZGMxZWNlNjE0ZDE5MjgxZDVkZGI2MGU4ZGE0MjllYzg5OTFmZDMzZjE0Mjg2OTY5MjM1OnQ6VA The funder had no role in the study design, data collection and analysis, decision to publish, or preparation of the manuscript. The contents of this manuscript are the responsibility of the authors and do not necessarily reflect the views of USAID or the US Government.”

Author reply: Thank you, please see our reply to #3 immediately above.

Author reply: We have revised our methods section, and removed this statement as it is no longer relevant.

6. Please include captions for your Supporting Information files at the end of your manuscript, and update any in-text citations to match accordingly. Please see our Supporting Information guidelines for more information: https://url.avanan.click/v2/___http://journals.plos.org/plosone/s/supporting-information___.YXAzOmlhdmk6YTpvOjRjOWRmZTI2ZWU3MTI1YjIxNjA2OGY0N2IwNzcxZWY5OjY6NDkwZjo1NzhhMGVkMzI1ZjgyZGVlNzRiOWI4ZWQ1Yjg1YTcyMTAwNTllMDgwOWZkMGFhMDM0MGYzMWYzY2YyYzk5MGY1OnQ6VA.

Author reply: I presume this doesn’t apply to our manuscript; we include no supplemental / supporting files or data at the end of our manuscript.

Additional Editor Comments (if provided):

Major comments:

1. The title of the paper and the objectives described are not sufficiently relatable. The title has a focus on STI burden whilst the aim described in the paper is looking at the feasibility and acceptability of recruiting and enrolling adolescents. The results presented are thus unrelated to the objectives set and thus cannot be thoroughly evaluated.

Author reply: Thank you for this comment, we have revisited our objectives (last paragraph of the introduction) to better match the data we present here. This is the first manuscript in a series of related publications from this study, some of the other papers will go into greater depth on topics more directly related to strategies to recruit and retain younger study participants. 

2. Authors must clearly describe the study setting. Initially (line 76) this is defined as, " Aurum Institute’s Clinical Research site in Rustenburg, Northwest Province." However, line 82-83 then lists additional study sites.

Author reply: We have clarified our methods (lines 87-88). In brief, the study was conducted at our research clinic, while recruitment, information sessions, and awareness raising was conducted at those locations noted in lines 82+

3. By restricting participants eligible to enrol for the study (pregnant, not willing to use contraceptive, diagnosis of HIV) the external validity of the results obtained is significantly reduced. Authors must then define to which population these results are generalizable.

Author reply: This is a good point. Our eligibility criteria were chosen to mimic what might be employed for a future clinical trial, e.g., an HIV vaccine trial, and was not necessarily intended to be generalized to a wider population than that. We have added a statement to that effect in our “limitations” paragraph (line 358). 

Minor comments

1. The sentences in line 46 - 47 need references and those in line 48 need references that are more recent and not from 2018 (these are already 5 years old)

Author reply: Thank you for catching this omission. We have added references where indicated. 

2. line 64: Define IAVI first before using the acronym.

Author reply: IAVI is no longer an acronym, it is the name of our organization (akin to KAVI, a partner organization that is also no longer an acronym). We have made note of this.

3. line 125: should read, viral "load"

Author reply: Thank you for catching this typo. We have corrected it per your comment. 

4. Please be consistent with terminology. The phrases, "young people, young women or young men" are not interchangeable with the word, "adolescent."

Author reply: You are correct, though in the context of our study, all participants were adolescents, as defined by the WHO and others. However, we have tried to be careful when using those two terms, and refer to “adolescents” as all study participants, and “young women/men” as study participants/adolescents who are ages 18 and 19. We only referred to “young women/men” once (line 184), and I was unable to find any examples where this might be unclear; if you’re able to provide specific lines, we can revisit this, but at the moment, I have not made any edits. 

Reviewer #1: The authors report on the high prevalence and incidence of sexually transmitted infections among adolescents from one clinic in South Africa. The authors also show that with the provision of youth friendly services, enrolment and retention of adolescents in studies related to sexual health can be very good. I only have a few minor comments. Can the authors comment on the reasons why there were more female than male participants. 

Author reply: Thank you for your comment. We had hoped to have roughly equivalent enrollment, between males and females, but more females came for screening. Due to time constraints, we were not able to extend enrollment to allow for more males to join the study. We have made a note of this in the discussion section, under limitations. 

Also discuss the reasons why females had a higher prevalence and incidence of STIs than males.

Author reply: It’s well documented in the literature that STI and HIV rates are higher, sometimes by quite a lot, among AGYW compared to their male counterparts. We discuss this in the second paragraph starting on line 267 in the Discussion. No further edits have been made in regard to this comment. 

Can you also describe in more detail what makes the setting youth-friendly?

Author reply: We adopted South African Department of Health standards and guidelines for working with and providing services to adolescents. We recommend the reviewer revisit lines 100-111 for additional details. We haven’t added any additional details in our resubmission. 

Lines 46-47 add reference to the statement

Author reply: We have added references to support the first sentence in the introduction. 

Line 58 clarify what is meant by simulated vaccine trials

Author reply: I have added a comment to clarify what we mean by this. 

Line 76 can the authors provide some data with regards to the catchment population size for the clinic/ adolescent population size

Author reply: We have updated our methods section with additional details

Line 119 Given that adolescents may be reluctant to talk about issues relating to sexual health with their parents, can the authors comment on how this was handled in the study?

Author reply: This is always a sensitive and sometimes challenging topic! Adolescents and youth were not seen together with their parents/guardians (see line 119, see also the “limitations” paragraph in the discussion, around line 355). This was also a topic that was addressed in the focus group discussions, and will be published soon. We don’t present any data from the parent/guardian group in this paper. 

Line 133: Neisseria and Chlamydia are organism names and should be capitalised and written in italics

Author reply: Thank you, we have made this change. 

Table 3: please also include any STI as a category

Author reply: We have added this to Table 3. 

Reviewer #2: In this prospective observational cohort study, the authors report a high prevalence of incident sexually transmitted infections and pregnancy in adolescents in Rustenburg, South Africa. The authors describe an adolescent friendly service, effective methods of adolescent recruitment and high retention in this hard-to-reach population.

The data is well presented and tables are clear. The findings support previous literature reporting high rates of STIs in adolescents in Sub-Saharan Africa.

My main concern is regarding the general style of the scientific writing throughout the paper, it is often informal and inconcise. The first sentences of the discussion and conclusion are clear examples. Table headings should be reviewed and shortened (e.g table 3 ‘among everyone overall’), use of phrases such as ‘on the other hand’, ‘in a study mentioned above’ should be avoided.

Author reply: While I do appreciate that there are many ways to write a scientific manuscript, I must beg to differ with this reviewer’s comment. I have revisited the first paragraph of the discussion, and our conclusion paragraph, and respectfully disagree. I am also of the school of thought that Table and Figure legends should be comprehensive. I teach my students that if their Table or Figure was to fall to the ground by itself, that anyone could pick it up and know what was being presented because of clear and complete legends, titles, axis labels, etc. I do thank you for the comment, but I have not made any edits in response to this comment. 

These data are important but may be better suited to an STI focused journal, the authors are advised to review the scientific writing style.

Author reply: I do appreciate the comment, but we feel PLOS ONE is well suited to this manuscript (see for example the recent publication in PLOS ONE: Monteiro IP, Azzi CFG, Bilibio JP, Monteiro PS, Braga GC, et al. (2023) Prevalence of sexually transmissible infections in adolescents treated in a family planning outpatient clinic for adolescents in the western Amazon. PLOS ONE 18(6): e0287633. https://doi.org/10.1371/journal.pone.0287633). We have re-read and reviewed the manuscript, and have made modest edits throughout to tighten the language.

---

## [Editor Report · Decision Letter 1]

18 Jun 2024

PONE-D-23-24060R1High STI burden among a cohort of adolescents aged 12-19 years in a youth-friendly clinic in South AfricaPLOS ONE

Dear Dr. Price,

Thank you for submitting your manuscript to PLOS ONE. After careful consideration, we feel that it has merit but does not fully meet PLOS ONE’s publication criteria as it currently stands. Therefore, we invite you to submit a revised version of the manuscript that addresses the points raised during the review process.

**ACADEMIC EDITOR:** The term young women/men was used in the following lines: 97, 178, 266, 269. The notation needs to be consistent to read adolescents.

We look forward to receiving your revised manuscript.

Kind regards,

Tinashe Mudzviti, MPhil(MD)

Academic Editor

PLOS ONE
---

## [Editor Report · Decision Letter 2]

25 Jun 2024

High STI burden among a cohort of adolescents aged 12-19 years in a youth-friendly clinic in South Africa

PONE-D-23-24060R2

Dear Dr. Price,

We’re pleased to inform you that your manuscript has been judged scientifically suitable for publication and will be formally accepted for publication once it meets all outstanding technical requirements.

Kind regards,

Tinashe Mudzviti, MPhil(MD)

Academic Editor

PLOS ONE

---

## [Editor Report · Acceptance letter]

28 Jun 2024

PONE-D-23-24060R2 

PLOS ONE

Dear Dr. Price, 

I'm pleased to inform you that your manuscript has been deemed suitable for publication in PLOS ONE. Congratulations! Your manuscript is now being handed over to our production team.

Kind regards, 

on behalf of

Dr. Tinashe Mudzviti 

Academic Editor

PLOS ONE